# Research on a Sound Source Localization Method for UAV Detection Based on Improved Empirical Mode Decomposition

**DOI:** 10.3390/s24092701

**Published:** 2024-04-24

**Authors:** Tao Chen, Jiyan Yu, Zhengpeng Yang

**Affiliations:** School of Mechanical Engineering, Nanjing University of Science and Technology, Nanjing 210094, China; aolunsiqiu352229@163.com (T.C.); yzp_njust@njust.edu.cn (Z.Y.)

**Keywords:** sound source localization, EMD dynamic modal decomposition method, signal processing, feature extraction, Chan–Taylor algorithm

## Abstract

To address the challenge of accurately locating unmanned aerial vehicles (UAVs) in situations where radar tracking is not feasible and visual observation is difficult, this paper proposes an innovative acoustic source localization method based on improved Empirical Mode Decomposition (EMD) within an adaptive frequency window. In this study, the collected flight signals of UAVs undergo smoothing filtering. Additionally, Robust Empirical Mode Decomposition (REMD) is applied to decompose the signals into Intrinsic Mode Function (IMF) components for spectrum analysis. We introduce a sliding frequency window with adjustable bandwidth, which is automatically determined using a Grey Wolf Optimizer (GWO) with a sliding index. This window is used to lock and extract specific frequencies from the IMFs. Based on predefined criteria, the extracted IMF components are reconstructed, and trigger signal times are analyzed and recorded from these reconstructed IMFs. The time differences between sensor receptions are then calculated. Furthermore, this study introduces the Chan–Taylor localization algorithm based on weighted least squares. This advanced algorithm takes sensor time delay parameters as input and solves a set of nonlinear equations to determine the target’s location. Simulations and real-world signal tests are used to validate the robustness and performance of the proposed method. The results indicate that the localization error remains below 5% within a 15 m × 15 m measurement area. This provides an efficient and real-time method for detecting the location of small UAVs.

## 1. Introduction

In military, civilian, and anti-terrorism sectors, precise detection of micro-UAV locations is crucial, yet traditional detection methods are costly and complex to operate. Passive acoustic detection systems, especially those employing the Time Difference of Arrival (TDOA) algorithm, offer a low-cost and efficient solution to this challenge [1]. However, there are three major obstacles in practical applications: accuracy issues in low signal-to-noise ratio (SNR) environments, difficulties in solving nonlinear equation systems, and limitations in long-range detection.

To address the low SNR problem, this paper introduces the Robust Empirical Mode Decomposition (REMD) algorithm. It incorporates robustness criteria or regularization terms, such as the Self-Adaptive Sifting Stopping Criterion (SSSC). By improving the endpoint effects of traditional EMD [2], REMD significantly enhances the stability and accuracy of signal processing, especially in complex and noise-mixed signal environments. The REMD method can adapt to various complex signal characteristics and effectively avoid noise interference and outlier influences, resulting in more accurate and reliable reconstructed signals. This solves the defect of EMD, which performs well in processing nonlinear and non-stationary signals but may be affected when facing complex signals or noise.

To address the difficulty in solving nonlinear equation systems, we combine the advantages of the Chan algorithm and the Taylor algorithm, proposing a novel Chan–Taylor fusion algorithm [3]. This approach not only reduces the difficulty of localization but also significantly improves localization accuracy, especially in noisy environments. To expand the detection range and increase the significance of time delay differences, we adopt a three-element discrete distributed array. By pre-measuring the specific coordinates of the sensors and combining them with the time delay estimation values obtained through REMD processing, we can accurately determine the target location using the Chan–Taylor fusion algorithm.

The structure of this paper is arranged as follows: Firstly, we introduce the division of far and near sound fields and the sound source localization model of the three-element discrete array. Secondly, we elaborate on the REMD time delay analysis method and then we delve into the specific mathematical methods of the Chan–Taylor fusion algorithm. Afterward, we present the simulation results of the REMD and Chan–Taylor algorithms. Finally, we summarize the full text. Through this structured arrangement, we aim to provide readers with a comprehensive and in-depth understanding of the applications and advantages of passive acoustic detection systems in micro-UAV localization. Figure 1 is a general flowchart of our approach to the methodology explained in this paper.

## 2. Sound Source Localization Model

### 2.1. Delineation of Near-Field and Far-Field Models

Assume that d is the array spacing, $\lambda_{m}$ is the minimum wavelength, L is the distance from the sound source to the microphone, and D=nd is the size of the receiver array. The waveform of the sound source arriving at the microphone array at this point is considered to be a spherical waveform, which is consistent with the near-field model, and vice versa, for a plane wave, which is consistent with the far-field model, when $L<\frac{2D^{2}}{\lambda_{m}}$. The near-field model is suitable for close sound sources, but considering the amplitude difference and phase difference in sound waves, it can more accurately describe the characteristics of sound wave propagation, thus improving the positioning accuracy, while the far-field model is the opposite [4,5]. Therefore, the near-field model is chosen in this paper, and its applicable area is improved by increasing the array aperture.

The near-field model requires three or more microphones for localization, using a three-microphone uniform linear microphone array as an example shown in the Figure 2. The first microphone $m_{1}$ is chosen as the reference microphone, and $\tau_{12}$ and $\tau_{13}$ denote the time delay between the acoustic signal arriving at the first microphone $m_{1}$ and the second microphone $m_{2}$ and the third microphone $m_{3}$, respectively. Then, the time delay $\tau_{12}$ and $\tau_{13}$ can be expressed as:(1)$$\tau_{12}=\frac{r_{2}-r_{1}}{c}$$
(2)$$\tau_{13}=\frac{r_{3}-r_{1}}{c}$$

Note: $r_{i}(i=1,2,3)$ is the distance from the point source to the i microphone, and c is the velocity of the propagation of the acoustic signal in air (generally taken as an approximation of 340 m/s).

### 2.2. Discrete Distributed Microphone Arrays

The acoustic signals received by microphone arrays based on the near-field model are spherical waves with a high correlation among signals. By increasing the aperture of the array, the reception range of a spherical wave can be enlarged, i.e., the practical application area of the microphone array can be increased. Therefore, a discrete distributed microphone array is selected, as shown in Figure 3.

In the coordinate system Oxy, the position of the N microphones are $m_{1}$$\left(X_{1},Y_{1}\right)$, $m_{2}$$\left(X_{2},Y_{2}\right)$, $m_{3}$$\left(X_{3},Y_{3}\right)$$\cdot \cdot \cdot m_{n}$$\left(X_{n},Y_{n}\right)$. The array of N microphones can be flexibly arranged. $T\left(X,Y\right)$ is the true position of the point source, the distance from T to the origin of the Oxy coordinate system is the drop-off distance R, the distance to each microphone $m_{i}$ is $r_{i}$(i=1,2,3), and the time at which the signal arrives at the reference microphone $m_{1}$ is t. According to the geometrical relationship, the following can be obtained [6]:(3)$$\left\{\begin{array}{l} X^{2}+Y^{2}=R^{2}\\ {\left(X-X_{1}\right)}^{2}+{\left(Y-Y_{1}\right)}^{2}={r_{1}}^{2}={\left(t*c\right)}^{2}\\ {\left(X-X_{2}\right)}^{2}+{\left(Y-Y_{2}\right)}^{2}={r_{2}}^{2}={\left(R+\tau_{12}*c\right)}^{2}\\ {\left(X-X_{3}\right)}^{2}+{\left(Y-Y_{3}\right)}^{2}={r_{3}}^{2}={\left(R+\tau_{13}*c\right)}^{2}\\ \cdot \cdot \cdot \\ {\left(X-X_{n}\right)}^{2}+{\left(Y-Y_{n}\right)}^{2}={r_{n}}^{2}={\left(R+\tau_{1n}*c\right)}^{2} \end{array}\right.$$

From Formula (3), we can obtain the hyperbolic intersection equations for the discrete distributed microphone array. By solving this system of hyperbolic equations, we can obtain the location information of the target. This system of equations is nonlinear because it contains square roots and squared terms. To solve this system of equations, we usually need to use nonlinear optimization algorithms, such as Newton’s method, the gradient descent method, or the least squares method. Specifically, we can define an objective function, which is the sum of the squared differences between the left and right sides of all equations, as shown in Formula (4) [7].
(4)$$F(X_{s},Y_{s})=\sum_{n=2}^{N}(\sqrt{{\left(X_{s}-X_{n}\right)}^{2}+{\left(Y_{s}-Y_{n}\right)}^{2}}-\sqrt{{\left(X_{s}-X_{1}\right)}^{2}+{\left(Y_{s}-Y_{1}\right)}^{2}}-c*\tau_{1n}{)}^{2}$$

Our goal is to find the value of $\left(X_{s},Y_{s}\right)$ that minimizes $F(X_{s},Y_{s})$. Additionally, the selection of the reference microphone is not fixed and can be randomly chosen as the $i_{th}$ microphone, which can then be combined with the other N−1 microphones.

## 3. Time Delay Analysis of Sound Source Signals

### 3.1. Frequency Window Robust Empirical Mode Decomposition Method

#### 3.1.1. Empirical Mode Decomposition (EMD)

The Empirical Mode Decomposition method (EMD) is a method of signal decomposition based on the time scale characteristics of the data itself without any predefined basis functions. Theoretically, it can achieve time resolution and frequency resolution in any dimension, providing a new method for analyzing and guiding smooth and nonlinear signals [8]. Based on its own characteristics, EMD makes the method theoretically applicable to the decomposition of any type of signal, and it has very obvious advantages and high signal-to-noise ratios in dealing with non-smooth and nonlinear data [9].

The traditional EMD decomposition steps can be categorized as follows:
(1)Find all extreme points of the original signal.(2)Fit the upper and lower envelopes $e_{\max}(n)$ and $e_{\min}(n)$ by using cubic spline curves.(3)Find the mean of the upper and lower envelopes and draw the mean envelope $m_{i}(n)$.
(5)$$m_{i}(n)=\frac{e_{\max}(n)+e_{\min}(n)}{2}$$(4)Obtain the middle signal $h_{i}(n)$ as the original signal minus the mean envelope.



(6)
$$h_{i}(n)=x_{i}(n)-m_{i}(n)$$



During the generation of intermediate signals, a conditional judgement on $h_{i}(n)$ is required as follows:In the interval, the number of extreme points and the number of points past zero must be equal or differ by at most one.At any time, the average value of the upper envelope formed by the local maxima and the lower envelope formed by the local minima is zero.

When the above conditions are met, an IMF component can be generated, and the algorithm updates the original signal and repeats the process.
(7)$$c_{1}(n)=IMF_{1}(n)=h_{i}(n)$$
(8)$$r_{1}(n)=x(n)-c_{1}(n)$$

Note: $c_{1}(n)$ is the first-order IMF component and $r_{1}(n)$ denotes the updated signal.

#### 3.1.2. Robust Empirical Mode Decomposition with Adaptive Frequency Windows

This paper proposes the use of adaptive frequency windows for locking and extraction in the signal spectrum to solve the problem of missing spectrum boundaries when segmenting the original signal. The schematic process is shown in Figure 4. The frequency window shown in the figure is $\left[f_{a},f_{b}\right]$, where $f_{a},f_{b}$ are the center frequencies of the front and back stopbands of the window. The shaded area is the segmented fragment transition region with a width of 2π [10]. After initially setting a sliding window width, the sliding window is used to traverse the decomposition frequencies in sequence. The bandwidth range of the frequency window is variable. While traversing, the frequency amplitude within the sliding window $\left[f_{a},f_{b}\right]$ is solved in parallel. When the calculated amplitude exceeds the set threshold, the sliding window locates the conforming frequency band and outputs the traversal result data.

As the traditional EMD algorithm adopts cubic spline interpolation in the decomposition, the algorithm has endpoint effects when decomposing the IMF. This leads to signal distortion, and there is a large error in the decomposition of the obtained IMF time axis. The principle of the algorithm leads to modal aliasing phenomenon in the decomposed IMF. When the signal-to-noise ratio of the original signal is low, the signal-to-noise ratio of the decomposed IMF is also low, and the effect becomes larger in further decomposition. Since the UAV flight sound is still mixed with noise in different frequency intervals during the process of being received by the sensors, it is necessary to improve the traditional EMD algorithm to adapt to the scenarios applied in this paper.

Aiming at the phenomena existing in the traditional EMD, combined with the application requirements of sound source localization, this paper adopts the improved Empirical Mode Decomposition algorithm, i.e., Robust Empirical Mode Decomposition (REMD).The main idea of REMD is to reduce the influence of noise in the process of signal decomposition and to improve the stability and reliability of the decomposition through the introduction of some robustness criteria or regularization terms [11]. By adjusting the regularization parameter or robustness criteria in the algorithm, the influence of noise can be reduced while retaining the signal characteristics. When the adaptive frequency window locks and splits the original signal, the high and low frequency signals in the split signal are decomposed by weights, and the decomposed IMF components are reconstructed, so as to obtain the reconstructed signal with a high signal-to-noise ratio. The specific implementation process is shown in Figure 5.

An REMD decomposition of the segmented signal x(n) is performed, and the decomposition yields n IMF components.
(9)$$x(n)=\sum_{i=1}^{N}c_{i}(n)+r_{N}(n)$$

Note:$\;c_{i}(n)$ is the improved robust intrinsic modal function and $r_{N}(n)$ is the residual term.

1.Filter the components with higher signal-to-noise ratio according to the following reconstruction criterion:
(1)Correlation coefficientThe correlation coefficient is a statistical indicator of the close connection among response variables. In this paper, Pearson’s correlation coefficient, which is commonly used in statistics, is used [12]. The degree of correlation between X and Y is described by a value in the interval [−1, 1]; the larger the value of the coefficient, the higher the correlation, and vice versa. After performing REMD decomposition, the algorithm automatically calculates the correlation coefficients between the IMF components and the segmented signals as a way to differentiate the useful components and retain them. Let two samples be X and Y. The following equation represents the Pearson correlation coefficient (ρx,y) of two continuous variables (x,y), which is equal to the product of the covariance cov(X,Y) between them divided by their respective standard deviations $\left(\sigma_{x},\sigma_{y}\right)$. Variables close to 0 are considered uncorrelated and those close to 1 or −1 are said to be strongly correlated.
(10)$$\begin{array}{l} \rho_{X,Y}=\frac{\mathrm{cov}(X,Y)}{\sigma_{X}\sigma_{Y}}=\frac{E\left((X-\mu_{x})(Y-\mu_{y}\right)}{\sigma_{X}\sigma_{Y}}=\frac{E(XY)-E(X)E(Y)}{\sqrt{E(X^{2})-E^{2}(X)}\sqrt{E(Y^{2})-E^{2}(Y)}}\\ \rho_{X,Y}=\frac{N\sum XY-\sum X\sum Y}{\sqrt{N\sum X^{2}-{\left(\sum X\right)}^{2}}\sqrt{N\sum Y^{2}-{\left(\sum Y\right)}^{2}}}\\ \rho_{X,Y}=\frac{\sum XY-\frac{\sum X\sum Y}{N}}{\sqrt{\left(\sum X^{2}-\frac{{\left(\sum X\right)}^{2}}{N})(\sum Y^{2}-\frac{{\left(\sum Y\right)}^{2}}{N}\right)}} \end{array}$$(2)Amplitude ratioThe mathematical expression for the energy ratio coefficient is
(11)$$\varepsilon =\frac{E_{IMF}(i)}{E_{x}}$$
where $E_{x}$ is the total energy of the IMF component; $E_{IMF}(i)$ is the energy of different IMFs; and ε is the energy ratio coefficient.
2.According to the proportional weighting method, take the maximum correlation coefficient as $IMF_{\max}(n)$ and the minimum correlation coefficient as $IMF_{\min}(n)$, subtract the two components and compute the average value, add the value to the segmented signal x(n), and reconstruct it to obtain a signal with high signal-to-noise ratio.
(12)$$x’(n)=x(n)+\frac{IMF_{\max}(n)-IMF_{\min}(n)}{2}$$3.Analyze the trigger time values in the reconstructed signal.

#### 3.1.3. Time Delay Estimation Method Based on the Comparison of Amplitude Mean Values of Reconstructed Signals

The time delay estimation method based on the comparison of amplitude mean values of reconstructed signals involves calculating the amplitude α at each sampling point of a pure signal and then determining its average value $\bar{\alpha }$. By comparing the amplitude mean values of the reconstructed signal and the reference signal, the time delay of the signal can be inferred. If there is a shift in the amplitude mean of the reconstructed signal relative to the reference signal, this offset can serve as an estimate $\tau_{ij}$ for the time delay. The effectiveness of this approach relies on the characteristics of the signal and the noise level. However, the use of improved REMD (Robust Empirical Mode Decomposition) can circumvent this issue.

To simplify the calculation, Figure 5 illustrates that this study only focuses on the amplitude within the first peak of the reconstructed signal to calculate the mean amplitude value. One of the reconstructed REMD signals from a microphone is selected as the reference signal, and its mean amplitude is compared with that of other microphone signals. The offset of the same amplitude value is chosen as the time delay estimation. When the ratio of the mean amplitude value ${\bar{\alpha }}_{i}$, ${\bar{\alpha }}_{i-1}$ at time $T_{i}$ to that at time $T_{i-1}$ reaches its maximum, we consider that $\tau_{i,i-1}=T_{i}-T_{i-1}$.

### 3.2. GWO-Based Adaptive Optimization

The Grey Wolf Optimization (GWO) algorithm is inspired by grey wolf packs in nature, and the GWO algorithm simulates the hierarchy and hunting mechanism of grey wolf packs in nature [13]. The Grey Wolf Optimization algorithm does not require complex parameter tuning, and, at the same time, it has a strong global search ability, which can quickly find the optimal solution. This article seeks the optimal fitness function of the adaptive frequency window with the help of the advantages of this algorithm. GWO classifies the entire wolf population into Alpha, Beta, Delta, and Omega wolves according to the fitness curve, with wolves decreasing in rank in the following update process:(1)Calculate the distance between an individual grey wolf and its prey and update the grey wolf’s position at all times:
(13)$$D(t)=\left|C*X_{p}(t)-X(t)\right|$$
(14)$$X(t+1)=X_{p}(t)-A*D(t)$$
where t is the number of moment iterations; D(t) is the distance between the individual grey wolf and the prey; $X_{p}$ is the position vector of the prey; X is the position vector of the individual grey wolf; and A,C is the coefficient vector, which is calculated as:(15)$$A=2\alpha *r_{1}-\alpha$$
(16)$$C=2r_{2}$$
(17)$$\alpha =2(1-t/t_{\max})$$
where $r_{1},r_{2}$ is the random vector in the interval [0, 1], $t_{\max}$ is the maximum number of iterations, and a is the convergence factor.

Grey wolves are able to identify the location of their prey and surround them [14]. When a grey wolf identifies the location of a target, β and δ guide the pack to encircle the prey under the leadership of α. Constantly update their positions using their tracking mathematical model expressions as follows:(18)$$D_{\alpha }=\left|C_{1}*X_{\alpha }(t)-X(t)\right|$$
(19)$$D_{\beta }=\left|C_{2}*X_{\beta }(t)-X(t)\right|$$
(20)$$D_{\delta }=\left|C_{2}*X_{\delta }(t)-X(t)\right|$$
(21)$$X_{1}=X_{\alpha }-A_{1}*(D_{\alpha })$$
(22)$$X_{2}=X_{\beta }-A_{2}*(D_{\beta })$$
(23)$$X_{3}=X_{\delta }-A_{3}*(D_{\delta })$$
(24)$$X(t+1)=\frac{X_{1}+X_{2}+X_{3}}{3}$$
where $D_{\alpha },D_{\beta }$ and $D_{\delta }$ denote the distance between α,β and δ and other individuals, respectively; $X_{\alpha },X_{\beta }$ and $X_{\delta }$ represent the current positions of α,β and δ, respectively; $C_{1},C_{2}$ and $C_{3}$ are random vectors; and X is the current position of the grey wolf. Equations (21)–(23) denote the step length and direction of individual w in the wolf pack towards e, f, and g, respectively, and Equation (24) denotes the final position of W.

The coefficient vector A in Equation (15) represents a random value in the interval $\left[-\alpha ,\alpha \right]$, and during the attack process of the wolf pack, the value of a is gradually reduced, while the range of fluctuation in A is also reduced. When $\left|A\right|<1$, the wolf pack approaches and captures the prey, i.e., the optimal solution is obtained, and when $\left|A\right|>1$, the grey wolf separates from the prey and searches for a more suitable target [15]. In this paper, the noise ratio in the frequency domain is used as the fitness function of the frequency window, and the GWO algorithm is used to adaptively determine the window position. Figure 6 shows the schematic of the GWO adaptive approach.

## 4. Improved Sound Source Localization Algorithms

### 4.1. TDOA Algorithm

The TDOA algorithm works by measuring the time differences between signals received by different receivers and uses these time differences to calculate the position of the transmitting source. Specifically, it is assumed that there are three or more receivers with known relative positions to each other and that when the source transmits a signal, each receiver records the time at which the signal is received. Since the velocity of signal propagation is known, this time difference and velocity information can be used to construct a system of localization hyperbolic equations to calculate the position of the signal source. However, this hyperbolic system of equations tends to be nonlinear, and the obtained solution is often imaginary, or there is no solution, which needs to be solved by transforming the nonlinear system of equations into a linear system of equations. The mainstream methods are the Fang algorithm, the Chan algorithm, and the Taylor algorithm.

In this paper, the Chan algorithm and the Taylor algorithm are selected, and five different sizes of Gaussian white noise are selected for simulation. Figure 7 represent the Chan algorithm measurement noise standard deviation, the Taylor algorithm measurement noise standard deviation, and the Chan and Taylor algorithm measurement noise standard deviation, respectively.

The simulation results in Figure 7 show that the RMSE (Root Mean Square Error) of the Chan algorithm and the Taylor algorithm are basically the same when the noise is small. As the noise of the measurements becomes bigger and bigger, the Taylor algorithm’s RMSE is obviously better than that of the Chan algorithm.

### 4.2. Improving the Chan–Taylor Localization Algorithm

As the Chan algorithm can make full use of the information of the TDOA measurement value of each base station, it can effectively reduce the negative impact of the large measurement error of individual base stations on the positioning, and the Taylor algorithm obtains higher positioning accuracy when the initial position selection accuracy is higher. Therefore, in this paper, the Chan algorithm and the Taylor algorithm are fused to locate the position of experimental micro-UAV, and five different sizes of Gaussian white noise are selected for simulation.

The principle of the Chan–Taylor joint positioning algorithm is to use the Chan algorithm to estimate the initial coordinates of the target $\left(x_{0},y_{0}\right)$. After the Chan algorithm, Taylor iterative calculations are carried out to constantly update the position label obtained by the method that more closely resembles the true position $\left(x,y\right)$. In this paper, the Chan–Taylor algorithm is selected and combined with the weighted least squares method for solving. The detailed derivation process is shown in Appendix B; here, we simply show the algorithm processing steps.

(1)In the first step, apply WLS estimation to the nonlinear equations, to obtain a quadratic equation in terms of $r_{1}$

.
(25)$$a*{r_{1}}^{2}+b*r_{1}+c=0$$

Then, solve the equation for the two roots available, omit the invalid solutions, and substitute the valid solutions back into Equation (12) to obtain the estimated coordinates $\left(x_{0},y_{0}\right)$ of the location point T.

(2)Use the estimated coordinates $\left(x_{0},y_{0}\right)$ obtained by the Chan algorithm as the initial values of Taylor’s series expansion method. Performing a Taylor expansion of $f_{i}(x,y,x_{i},y_{i})$ at $\left(x_{0},y_{0}\right)$ and translate into matrix form:



(26)
$$\psi =h_{i}-G_{i}\delta$$



Finally, solve the weighted least squares solution of the above equation to obtain the final target coordinates:(27)$$\delta =\left[\begin{array}{l} \varepsilon_{x}\\ \varepsilon_{y} \end{array}\right]={\left(G_{i}^{T}Q^{-1}G_{i}\right)}^{-1}G_{i}^{T}Q^{-1}h_{i}$$
where $T\left(x,y\right)$ is the final positioning coordinate of the target that is infinitely close to the true value after iterative calculation.

## 5. Verification and Analysis

In this section, we test the performance and robustness of the proposed localization algorithm. In Section 5.1, we compare the EMD with the improved algorithm proposed in this paper by loading simulated signals under ideal conditions. We draw conclusions by comparing the IMF curves of the two algorithms. Section 5.2 verifies the actual performance of the strategy and algorithm proposed in this paper using signals collected in a real environment, thus demonstrating the accuracy and feasibility of the entire sound source localization algorithm proposed in this paper for drone localization scenarios.

### 5.1. Algorithm Stability Validation

Based on the analysis of the flight characteristics of drones, the signal generated by the rotating blades during drone acceleration can be regarded as a short-term impulse signal. This signal is characterized by a peak value appearing after a period of time, which gradually weakens over time. Because of the complex surrounding environment of the experiment, the drone flight signal may be interfered with by certain high-frequency vibration signals in the air and soil. As a result, the received signal may contain high-energy high-frequency vibration signals and low-amplitude random white noise signals. Therefore, we created a similar environment in the laboratory, where the hardware equipment consists of a DAQ122 acquisition card and a discrete microphone array composed of three MEMS digital microphones. The signal collected by the host computer is shown in the figure. We compared the EMD algorithm with the proposed algorithm using this analog signal. The simulation was performed using Matlab (Version: R2022b, created by MathWorks in Natick, MA, USA), and the simulation results are shown in Figure 8, Figure 9, Figure 10 and Figure 11.

The IMF diagrams decomposed by the two algorithms are both listed in Appendix A. By comparing the results in the appendix, it is concluded that the orthogonality between each IMF obtained after REMD of the collected signals is higher than that of the IMF decomposed by EMD, which makes the decomposed IMF more independent. At the same time, after improving robustness, the IMF decomposed by REMD fluctuates significantly, and the IMF decomposed at low frequencies has more convergent stability. There is no signal aliasing in each IMF, which can more accurately capture the local characteristics of the signal, thus greatly improving the performance of later signal reconstruction and adaptive sliding window positioning.

### 5.2. Real Signal Simulation Experiment

Within the test area, to eliminate the influence of human manipulation on the drone’s flight position, a system verification test was conducted using pre-recorded audio of a drone’s accelerating ascent sound signal to simulate drone flight. In the experiment, the actual positions of the sensors were preset and further corrected using differential GPS for positional coordinates. The recorded drone flight sound signal was then amplified and played at a designated point. The microphone array in the test site collected and processed these signals. After obtaining the time delay values, the TDOA algorithm was utilized for positioning. The system recorded each solution result and compared the positioning outcomes with the preset real target points to evaluate positioning accuracy and system robustness. Figure 12 shows the microphone placement at the test site.

A total of 10 valid tests were conducted in the experiment. Figure 13 shows a schematic diagram of the target positions and sensor positions corresponding to the ten tests. The same drone flight sound signal was used at all playback points, and the signal decibel was unified. Figure 14 shows the pre-collected drone flight sound signals collected by three microphone sensors in the first experiment. According to pre-experiment tests, the signals collected by the sensors in the experiment were basically consistent with the pre-collected drone flight signals. Therefore, they could be used as experimental simulation signals. Meanwhile, there were slight environmental noises such as wind, people walking, and vehicles driving around during the experiment. Table 1 is a record of the trigger times for the signals collected by the system analysis, and Table 2 is a record of the ten tests for the real landing point (x,y) and the system positioning estimate landing point $\left(x^{*},y^{*}\right)$, which are used to calculate the positioning error δ.

As can be seen in Figure 14, because of the limited distance of the microphones, the time received by each microphone is very close. For the positioning algorithm adopted in this paper, the signals processed without any treatment or using traditional signal decomposition algorithms cannot guarantee the positioning accuracy of the drone. Next, we decompose the signals collected by the three microphones.

As shown in Figure 15, the amplitudes of the IMFs decomposed by the algorithm in this paper for the three microphones are the same, indicating that the filtering approach adopted in this paper completely separates the mixed noise in the acoustic wave. It can also be seen that the filtering used does not clip the useful signal at the top or bottom. The signal spectrum obtained by FFT transformation is shown in Figure 16. A sliding frequency window with variable bandwidth is used for corresponding frequency matching and extraction. The reconstructed signal filtered by the screening criteria is shown in Figure 17. By analyzing the data and using the algorithm to calculate the time difference between different sensor receptions, the trigger signal moments in the extracted IMF components are determined and recorded.

Following this process, the time delay results obtained from the remaining nine experiments are recorded in Table 1. Meanwhile, the estimated position coordinates are compared with the actual measured coordinates using the positioning algorithm presented in this paper, and the calculated error values are recorded in Table 2.

In Table 1 and Table 2, it can be seen that in the same experimental environment, after analysis using the improved Empirical Mode Decomposition sound source localization algorithm with an adaptive frequency window, the positioning estimation error is below 5% within the range where signals can be collected. This indicates that using the algorithm presented in this paper, the time delay estimation value resolved from the decomposed signal greatly improves the success rate of position estimation. This algorithm performs well in both near and far fields and even in the presence of random noise. However, during the experiment, it was found that the positioning accuracy was still not high enough in some locations. The specific reason is that because of product packaging and other factors, the omnidirectional microphone does not have exactly the same sensitivity in all 360-degree directions, which interferes with the selection of relevant points for time delay estimation. Thus, subsequent algorithms still need to make corresponding compensations for special values.

## 6. Conclusions

In this paper, a drone sound source localization system based on the Empirical Mode Decomposition (EMD) algorithm is constructed. In the presence of noise aliasing, the system can separate signals of different frequency bands and recombine them. Using frequency domain analysis of drone flight signals before the experiment, the complete signal of the drone flight can be effectively decomposed, thus improving the accuracy of time delay estimation. This also improves the rapid performance degradation of the Chan–Taylor algorithm when the time delay estimation accuracy is insufficient. At the same time, a distributed sensor array is adopted, which can be freely arranged, and a mathematical geometric positioning model for random arrangement of N microphones is derived, expanding the positioning area and flexibility.

However, this paper did not conduct experiments on real drone flights in actual test sites. Instead, the test conditions were carried out in a semi-open field environment without multipath interference. Our next research goal is to further improve the performance of the EMD algorithm when drone flight signals are reflected and aliased multiple times.

## Figures and Tables

**Figure 1 sensors-24-02701-f001:**
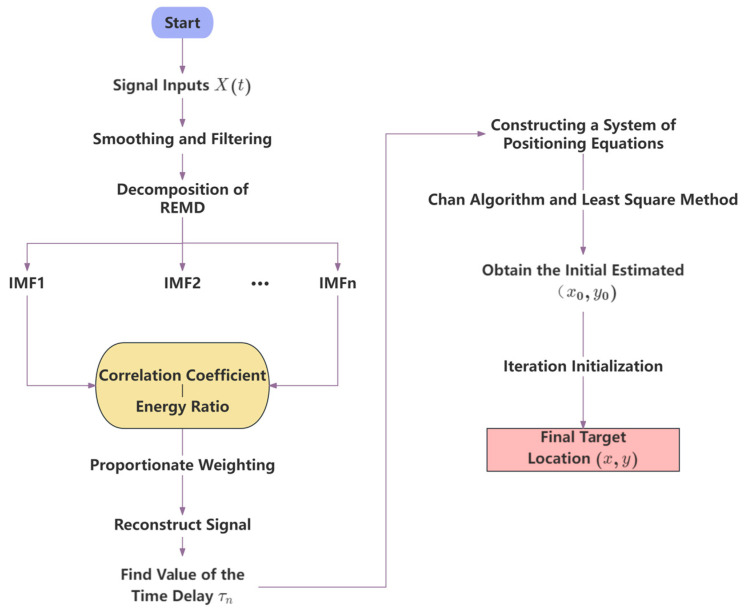
Positioning flowchart.

**Figure 2 sensors-24-02701-f002:**
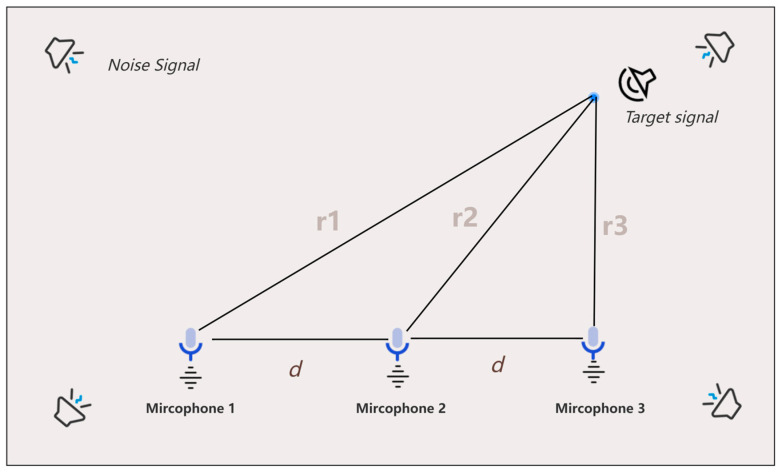
Near-field modeling of uniform line arrays.

**Figure 3 sensors-24-02701-f003:**
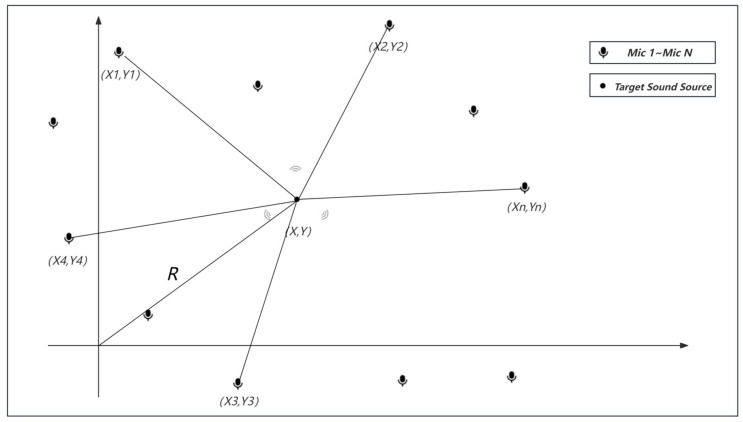
Discrete distributed microphone array model.

**Figure 4 sensors-24-02701-f004:**
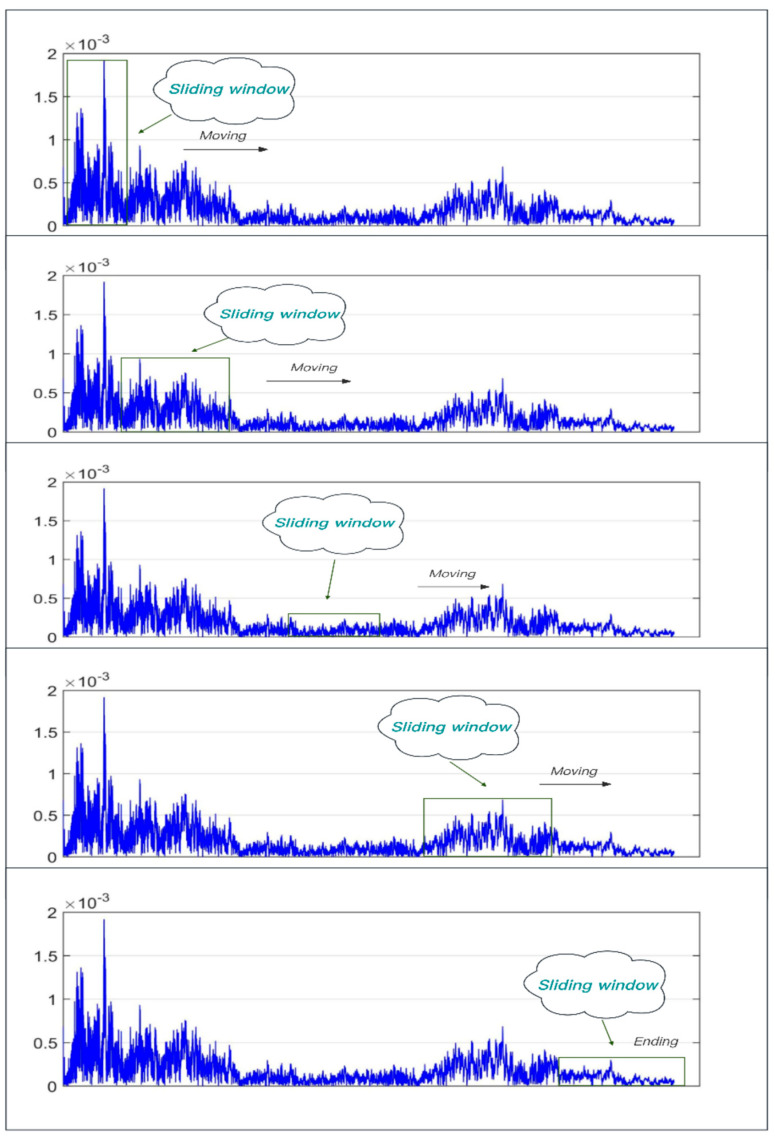
Adaptive frequency window.

**Figure 5 sensors-24-02701-f005:**
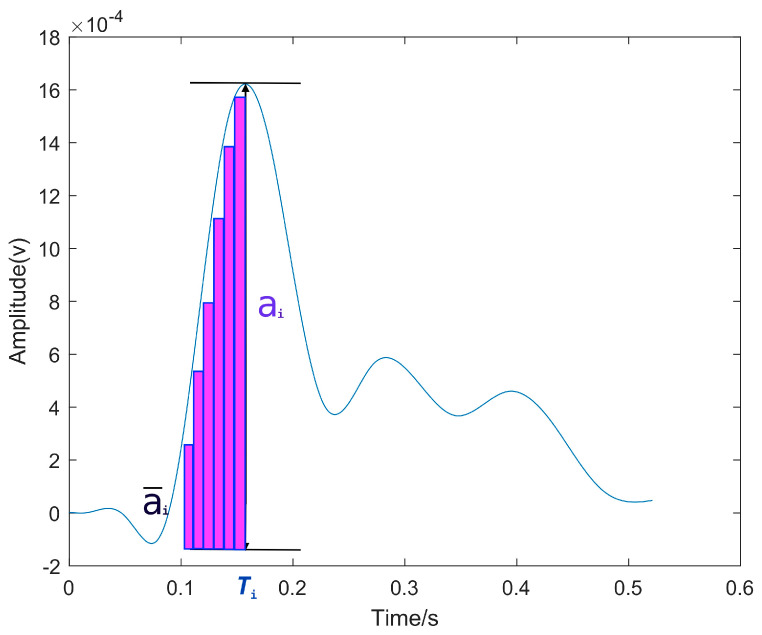
Schematic diagram of the time delay estimation method based on amplitude mean value comparison.

**Figure 6 sensors-24-02701-f006:**
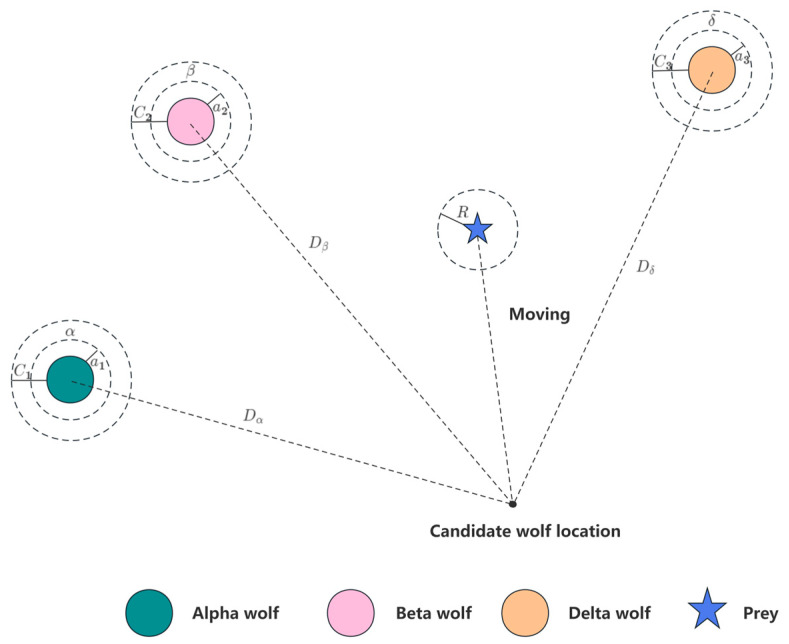
Schematic diagram of the GWO adaptive algorithm.

**Figure 7 sensors-24-02701-f007:**
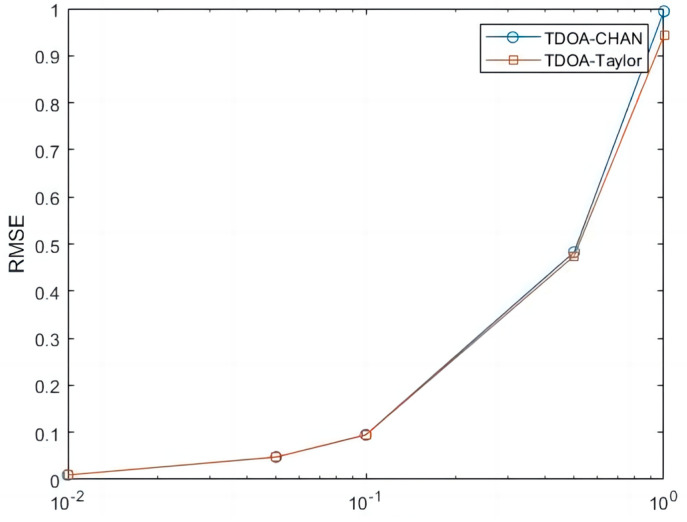
Different algorithms used to measure standard deviation.

**Figure 8 sensors-24-02701-f008:**
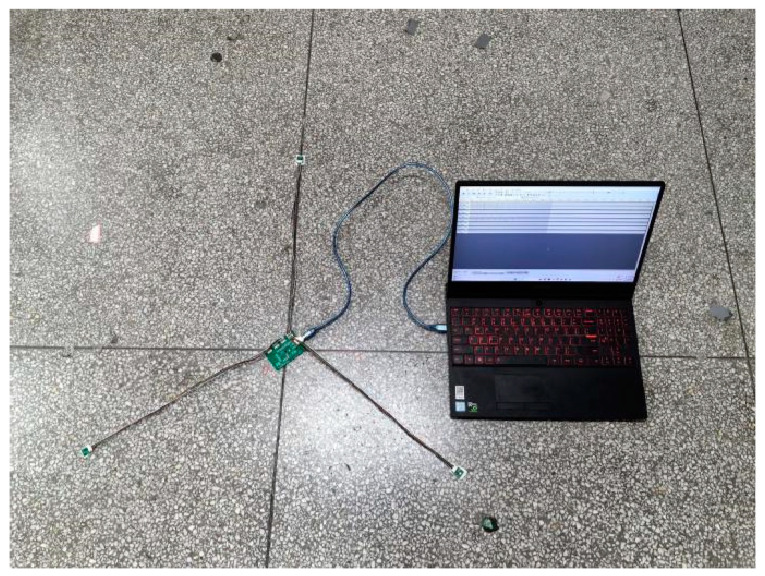
Site layout.

**Figure 9 sensors-24-02701-f009:**
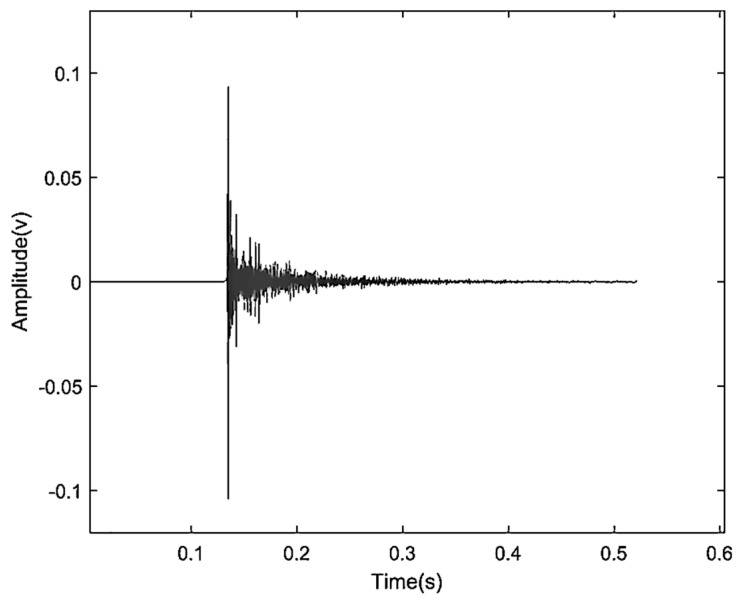
Simulated acoustic signal time domain diagram.

**Figure 10 sensors-24-02701-f010:**
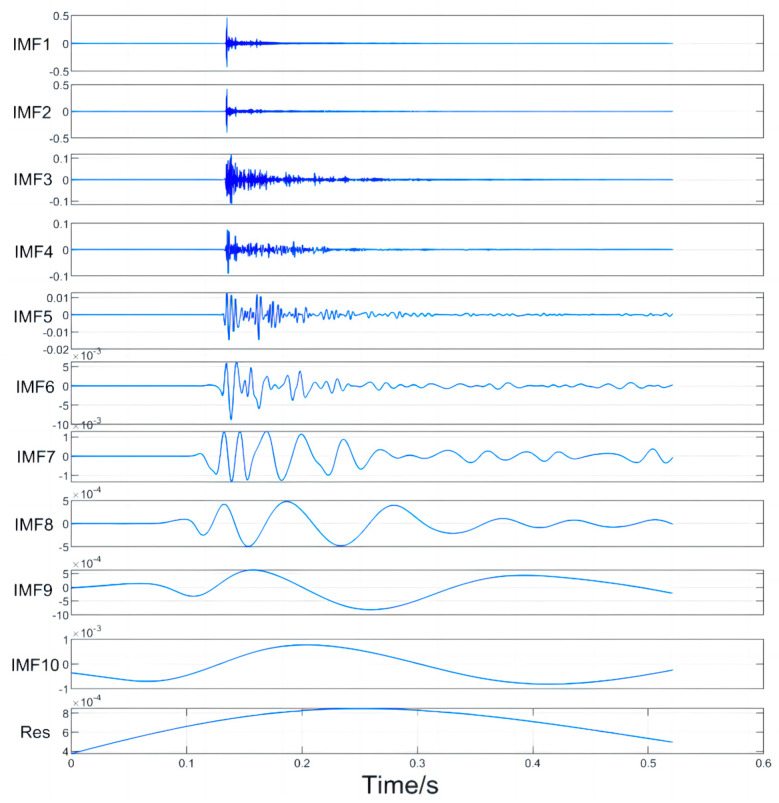
IMF decomposition of the EMD algorithm for simulating acoustic signals.

**Figure 11 sensors-24-02701-f011:**
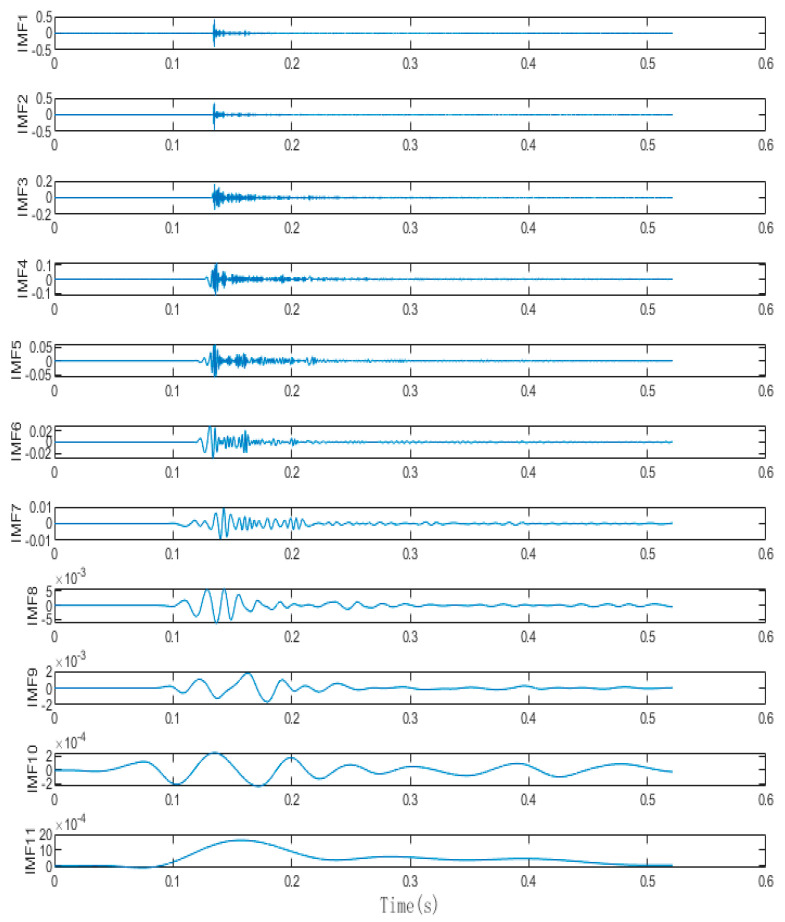
IMF decomposition of the REMD algorithm for simulating acoustic signals.

**Figure 12 sensors-24-02701-f012:**
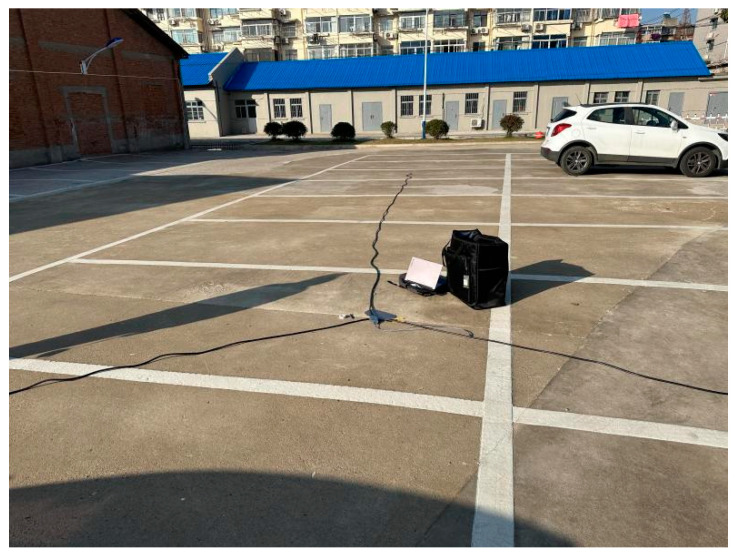
Map of sensor positions during the test.

**Figure 13 sensors-24-02701-f013:**
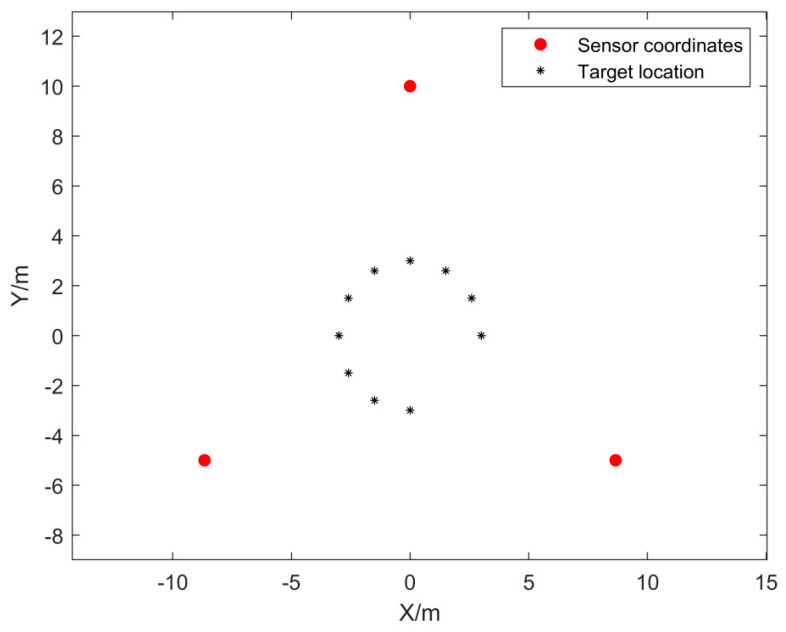
Schematic diagram of the corresponding target points and sensor positions in the test.

**Figure 14 sensors-24-02701-f014:**
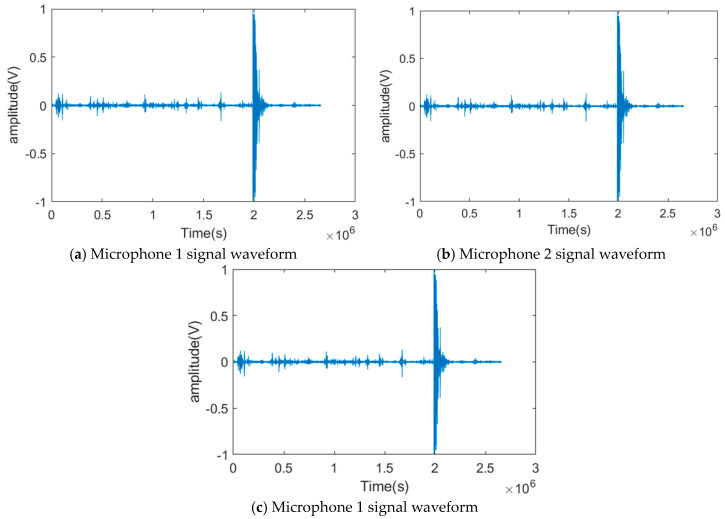
Waveforms of the signals collected by the three array microphones.

**Figure 15 sensors-24-02701-f015:**
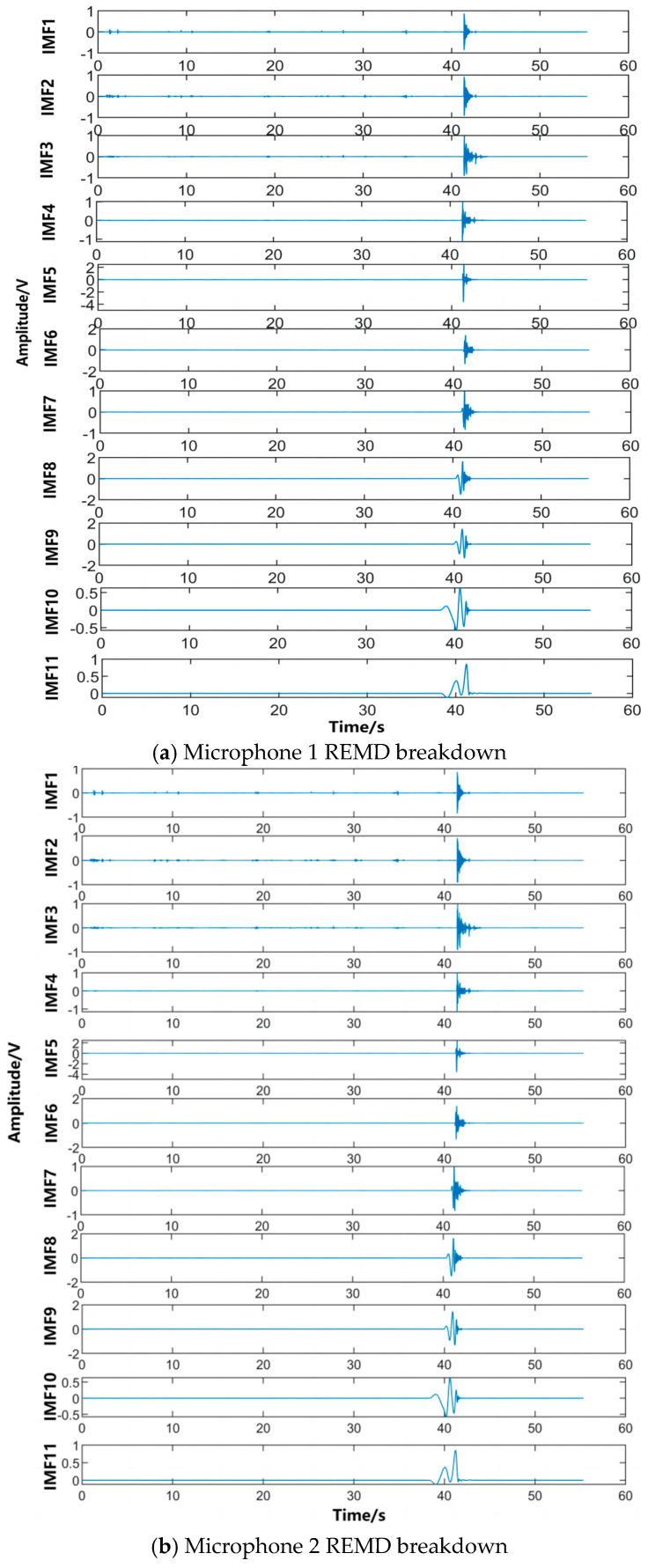
REMD decomposition of the signals collected by the three array microphones.

**Figure 16 sensors-24-02701-f016:**
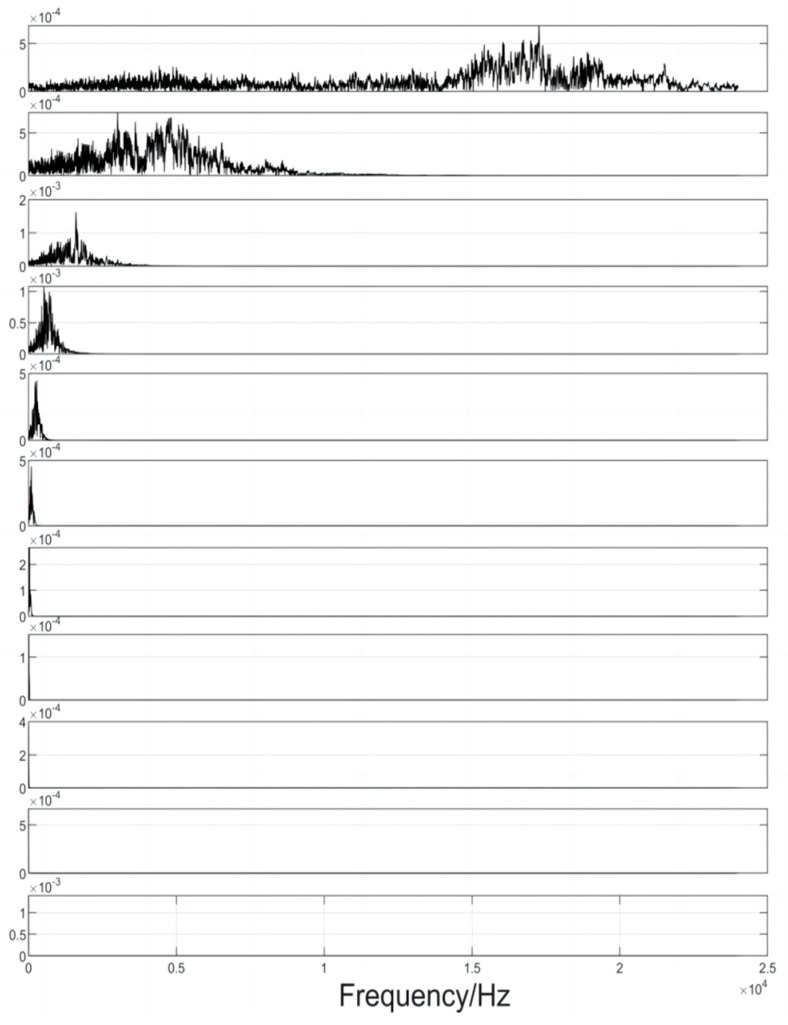
Spectrogram.

**Figure 17 sensors-24-02701-f017:**
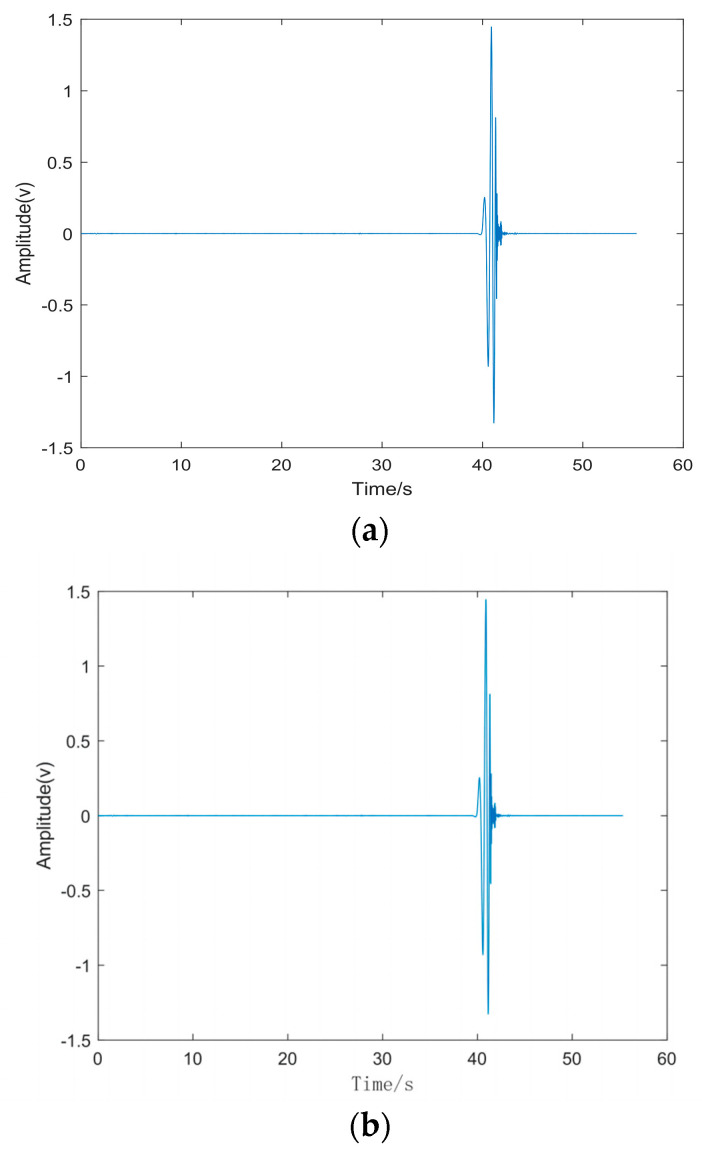
Reconstruction of the IMF trigger signal moment chart: (**a**–**c**) denote the IMF decomposition process of the signal, respectively, and c is the final IMF decomposition result.

**Table 1 sensors-24-02701-t001:** Time estimation results after REMD decomposition.

Serial Number	Mic(i) Trigger Moment/s			Mic2, Mic3, and Mic1 Time Difference/s	
	1	2	3	α_1_	α_2_
1	1.7123	1.7217	1.7282	0.00941	0.01592
2	1.6272	1.63073	1.64121	0.00353	0.01401
3	1.7233	1.73083	1.73851	0.00753	0.01521
4	1.2341	1.24808	1.24808	0.01398	0.01398
5	1.1752	0.18972	1.181631	0.01452	0.006431
6	2.3812	2.39351	2.38131	0.01231	0.00011
7	1.4251	1.43207	1.41696	0.00697	−0.00814
8	2.3127	2.3128	2.29879	0.0001	−0.01391
9	1.4527	1.44636	1.33037	−0.00634	−0.01498
10	1.8162	1.80387	1.80387	−0.01233	−0.01233

**Table 2 sensors-24-02701-t002:** Test drop monitoring results.

Serial Number	True Point of Impact		The Estimated Position		δ/%
	x/m	y/m	$x^{*}/m$	$y^{*}/m$	
1	3	0	2.9935	0.0098	4.48
2	2.598	1.5	2.61636	1.51643	2.41
3	1.5	2.598	1.49853	2.61034	1.00
4	0	3	0	3.02133	2.13
5	−1.5	2.598	−1.48913	2.62274	1.61
6	−2.598	1.5	−2.59035	1.49564	0.88
7	−3	0	−3.02718	0.00168632	2.72
8	−2.598	−1.5	−2.59451	−1.49775	0.41
9	−1.5	−2.598	−1.46837	−2.75018	9.77
10	0	−3	−0.0001569	−3.06854	4.85

## Data Availability

Data on supporting conclusions have been charted and presented in the article.

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
