# Peer review of "Research on a Sound Source Localization Method for UAV Detection Based on Improved Empirical Mode Decomposition"

_sensors, 2024, doi:10.3390/s24092701_

Round 1

Reviewer 1 Report

Comments and Suggestions for Authors

The sound Source localization method for UAV method with improved empirical modal decomposition under an adaptive frequency window is proposed in this paper. Some parts of the manuscript are not clearly written, and there are some issues that need further revision and explanatory notes.
1) Authors analyze three problems of the current sound source location system, focusing on the low signal-to-noise ratio. The experimental part should include relevant data results, which seems more reasonable to explain that the improved EMD is effective in the situation with low signal-to-noise ratio.
2) The improved EMD algorithm can be illustrated the advantages in time estimation in comparison with the EMD without improved in data.
3) Test drop monitoring results have been shown in table 2, and the serial number 9 is shown that the estimation error is nearly 10%. Authors should have to explain and discuss some possible reasons.

Reviewer 2 Report

Comments and Suggestions for Authors

The paper describes an UAV detection sound source positioning system based on an EMD improved algorithm, that is based on the robust empirical modal decomposition with an adaptive frequency window. The idea is interesting but the way that the paper was written does not help. The paper is very messy and difficult to understand. At this point, in my opinion, it is very hard to judge if the paper should be considered for publication. The abstract should completely rewritten! There are repeated text and the use of long sentences doesn’t help. This problem continues in the remaining document. The quality of the document is also bad:

1.) the equation font size is to big, the equation numbers are not aling; the * instead of × for product; 

2.) Figures numbering format is wrong and their quality/resolution is very low ;

3.) The document is to big… part of the information (results plots) could go to an appendix 

4.) Section 5 is to confuse, considering rewriting it…

5.) the bibliography should be revised…

Additionally, I recommend to rewrite the Introduction section, including a compressive revision of the state of the art. Moreover, the near-field discussion could be summarize to a single sentence this a well establish assumption in acoustic location systems using TDOA. Nonetheless in line 122: In my opinion d in the near-field equation (Fraunhofer distance) is not the distance between mics but the “size” of the receiver array. Another point that I would like to see discussed in the document is the use of only 3 microphones! Why not use more? The theoretical analysis should be generalized to N≥3, and therefore TDOA measurements from the target can be obtained by calculating the time difference of arrival using each combination of receiver and avoid the use of a mic 1 as reference. Another important point is section 3.1.2 I totally miss the Adaptive Frequency Windows explanation. I couldn't understand how it works… As it is the most important and innovative part of the paper It should be perfect! Finally some minor comments:

1.) What is the point of Figure 3.1a and b the information is in Figure 3.1c!

2.) Section 5.1 the figures reference in the text change to Fig. and the number is missing…
